# Postoperative Karnofsky performance status prediction in patients with *IDH* wild-type glioblastoma: A multimodal approach integrating clinical and deep imaging features

**Tomoki Sasagasako**[1,2], **Akihiko Ueda**[2], **Yohei Mineharu**[3]*, **Yusuke Mochizuki**[4], **Souichiro Doi**[4], **Silsu Park**[1], **Yukinori Terada**[1], **Noritaka Sano**[1], **Masahiro Tanji**[1], **Yoshiki Arakawa**[1], **Yasushi Okuno**[2,3]

**1** Department of Neurosurgery, Kyoto University Graduate School of Medicine, Kyoto, Japan, **2** Department of Biomedical Data Intelligence, Kyoto University Graduate School of Medicine, Kyoto, Japan, **3** Department of Artificial Intelligence in Healthcare and Medicine, Kyoto University Graduate School of Medicine, Kyoto, Japan, **4** Kyoto University Faculty of Medicine, Kyoto, Japan

* mineharu@kuhp.kyoto-u.ac.jp

**Data Availability Statement:** The code used to develop the model described herein is publicly

## Abstract

### Background and purpose

Glioblastoma is a highly aggressive brain tumor with limited survival that poses challenges in predicting patient outcomes. The Karnofsky Performance Status (KPS) score is a valuable tool for assessing patient functionality and contributes to the stratification of patients with poor prognoses. This study aimed to develop a 6-month postoperative KPS prediction model by combining clinical data with deep learning-based image features from pre- and postoperative MRI scans, offering enhanced personalized care for glioblastoma patients.

### Materials and methods

Using 1,476 MRI datasets from the Brain Tumor Segmentation Challenge 2020 public database, we pretrained two variational autoencoders (VAEs). Imaging features from the latent spaces of the VAEs were used for KPS prediction. Neural network-based KPS prediction models were developed to predict scores below 70 at 6 months postoperatively. In this retrospective single-center analysis, we incorporated clinical parameters and pre- and postoperative MRI images from 150 newly diagnosed IDH wild-type glioblastoma, divided into training (100 patients) and test (50 patients) sets. In training set, the performance of these models was evaluated using the area under the curve (AUC), calculated through fivefold cross-validation repeated 10 times. The final evaluation of the developed models assessed in the test set.

### Results

Among the 150 patients, 61 had 6-month postoperative KPS scores below 70 and 89 scored 70 or higher. We developed three models: a clinical-based model, an MRI-based model, and a multimodal model that incorporated both clinical parameters and MRI features. In the

available on GitHub: https://github.com/TomokiSasagasako/GBM_KPS_prediction.git The raw data in this study cannot be shared publicly as it includes identifiable information such as the name of the treatment hospital (Kyoto University Hospital), diagnostic details, age, gender, and clinical outcomes. The raw data are available upon reasonable request from the corresponding author or Kyoto University Graduate School and Faculty of Medicine, Ethics Committee via email (ethcom@kuhp.kyoto-u.ac.jp) or telephone (+81-75-753-4680).

**Funding:** This work was supported by the Ministry of Education, Culture, Sports, Science, and Technology (MEXT) under the RIKEN joint research and collaboration fund for "Translational Research in Basic and Clinical Sciences for the Construction of an AI Pharmaceutical Platform". The funders had no role in study design, data collection and analysis, decision to publish, or preparation of the manuscript.

**Competing interests:** The authors have declared that no competing interests exist.

**Abbreviations:** BraTS, Brain Tumor Segmentation challenge; IDH, isocitrate dehydrogenase; KPS, Karnofsky performance status; VAE, variational autoencoder.

training set, the mean AUC was 0.785±0.051 for the multimodal model, which was significantly higher than the AUCs of the clinical-based model (0.716±0.059, P = 0.038) using only clinical parameters and the MRI-based model (0.651±0.028, P<0.001) using only MRI features. In the test set, the multimodal model achieved an AUC of 0.810, outperforming the clinical-based (0.670) and MRI-based (0.650) models.

## Conclusion

The integration of MRI features extracted from VAEs with clinical parameters in the multimodal model substantially enhanced KPS prediction performance. This approach has the potential to improve prognostic prediction, paving the way for more personalized and effective treatments for patients with glioblastoma.

## Introduction

Glioblastoma is a highly malignant brain tumor with a median overall survival of approximately 15–18 months [1]. Despite numerous studies on treatment strategies, it often recurs rapidly, leading to a worsened functional prognosis.

Deep learning has increasingly been applied to detect, diagnose, and predict clinical outcomes in patients with glioblastoma [2]. However, it is still uncertain whether a radiomics approach using deep learning approach can enhance the prediction of clinical outcome or if pre- and postoperative MRI features can reliably predict prognosis.

The Karnofsky Performance Status (KPS) score stands as a robust independent predictor of clinical outcomes within diverse oncology populations affected by malignant tumors [3]. A KPS score of 70 indicates the patient can care for themselves but is unable to carry out daily activities. Among patients with glioblastoma, multiple studies have indicated a correlation between a KPS score of <70 and a poor prognosis [4]. Commonly, the 6-month postoperative KPS score is used for evaluating disease progression in glioblastoma clinical trials [5]. Predicting the postoperative KPS, along with the early identification of patients at risk of diminished KPS in the postoperative stage, could lead to improved counseling and more personalized clinical decision-making [6].

In this study, we developed a multimodal model using clinical parameters and brain MRI to stratify patients into prognostic groups based on their 6-month postoperative KPS. We utilized a deep learning approach to extract imaging features from pre- and postoperative MRI images and investigated their prognostic value.

## Materials and methods

The overall study process is illustrated in Fig 1. The proposed algorithm comprises two primary stages: 1) constructing variational autoencoders (VAEs) to extract the reduced latent features from the pre- and postoperative MRI images, and 2) developing a KPS prediction model by combining the patients' clinical parameters and extracted imaging features. The authors collected patients' data from April 2022 to July 2023.

### Ethics approval

The Ethics Committee of Kyoto University Hospital approved this study (R2088). Verbal informed consent was obtained from the study participants. Participants were informed before

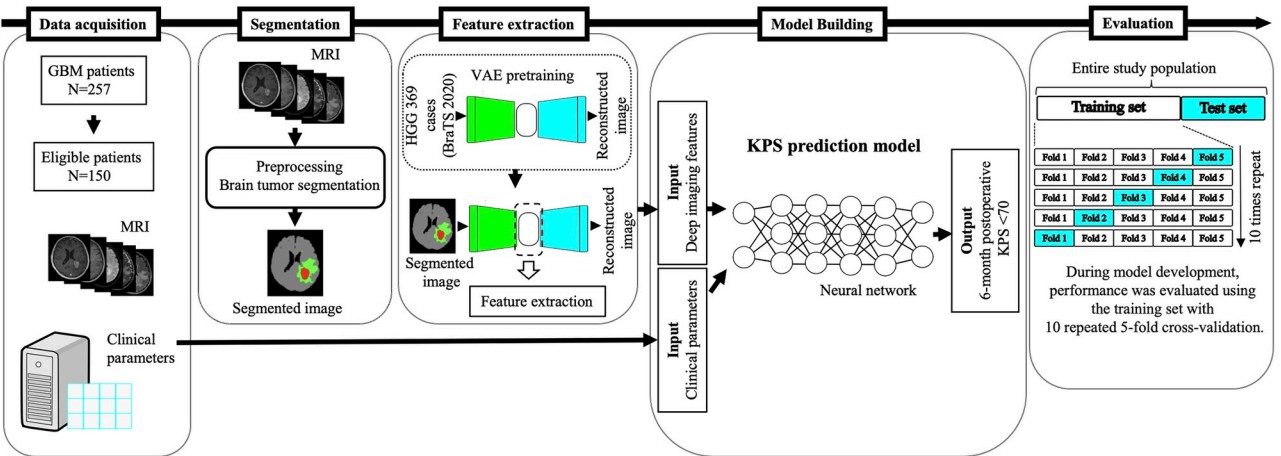

**Fig 1. Outline of the main steps in this study.** (1) Patient surveys and data acquisition from our institute's medical records, (2) brain tumor segmentation using pre- and postoperative MRI, (3) imaging feature extraction using a pretrained variation autoencoder combined with a convolutional neural network, and (4) prediction model development using a neural network and training. The performance metrics in the training set were evaluated using 10 repetition of fivefold cross-validation. GBM = glioblastoma; HGG = high-grade glioma; VAE = variational autoencoder.

their surgery that their images and clinical information would be used in a retrospective review after being fully anonymized. The ethics committee waived the requirement for written informed consent owing to the retrospective study design. Patients who did not wish to participate were excluded via an opt-out process.

## Patients

This single-center retrospective review was conducted between December 2001 and December 2022 at our institution. This study included consecutive adult patients aged 18 years and older who were newly histopathologically diagnosed with glioblastoma with isocitrate dehydrogenase (IDH) wild type. The exclusion criteria for the study cohort were as follows: 1) gliomas that were not histopathologically diagnosed as glioblastomas, 2) lack of immunohistochemical testing for the IDH1 R132H mutation or absence of IDH2 sequencing, 3) a diagnosis of IDH-mutant gliomas, and 4) a follow-up period of <6 months. A total of 257 patients were pathologically confirmed to have glioblastoma. Among these, 87 patients were excluded based on the following criteria: diagnosis of IDH-mutant gliomas (n = 11) or absence of either immunohistochemistry for the IDH1 R132H mutation or IDH2 sequencing (n = 76). Additionally, patients lacking sufficient medical records or imaging data (n = 18) and those aged <18 years (n = 2) were also excluded. Subsequently, patients were divided into training and test sets based on their first operation date. In studies with relatively small populations, the performance of a model on a test set can significantly vary with each random split due to substantial changes in the distribution of the test set data [7]. To mitigate the bias associated with arbitrarily selecting the test set through repeated random splits, we chose to divide the cases based on their time range into training and test sets in a 2:1 ratio, a method frequently employed in prior studies with similar cohort sizes [8, 9] Patients who underwent surgery from December 2001 to October 2018 were assigned to the training set, while those from November 2018 to December 2022 were included in the test set. This process resulted in 100 patients in the training set and 50 in the test set (S1 Fig).

## Clinical parameters and endpoints

According to previous research on machine-learning-based models predicting glioblastoma prognosis, the following 28 variables were incorporated as clinical parameters [10, 11].

Preoperative variables: sex, age at diagnosis, dominant hand (right/left), epilepsy (yes/no), aphasia (yes/no), paralysis (yes/no), other neurological findings at onset (yes/no), and preoperative KPS score (%).

Intraoperative variables: surgical strategy (biopsy or tumor removal), awake surgery (yes/no), utilization of 5-aminolevulinic acid (yes/no), photodynamic therapy (yes/no), carmustine wafer placement (yes/no), and motor-evoked or somatosensory-evoked potential monitoring (yes/no).

Immunohistochemical and genetic variables: O-6-methylguanine-DNA methyltransferase (MGMT) methylation (positive/negative), *TERTp* alteration (positive/negative), MIB-1 labeling index (%), and immunohistochemical staining for MGMT (positive/negative).

Postoperative variables: TMZ chemotherapy (yes/no), bevacizumab chemotherapy (yes/no), radiation dose (Gy), number of radiation fractionations (Fr).

Radiological findings: tumor laterality (right/left/bilateral), ependymal involvement (yes/no), midline shift (yes/no), corpus callosum invasion (yes/no), necrotic or cystic area evident on imaging (yes/no), and extent of resection (1–49%, 50–89%, 90–99%, 100%).

Additionally, the 6-month postoperative KPS (%) score, the main endpoint of this study, was collected.

For the standard concurrent chemoradiotherapy regimen, patients received fractionated focal radiation therapy with a cumulative dose of 60 Gy, accompanied by concomitant TMZ chemotherapy [12]. For elderly patients aged 70 years or older, a hypofractionated radiotherapy schedule of 40 Gy delivered in 15 fractions over 3 weeks was employed [13]. All patients were followed up every 1–2 months after surgical treatment.

## Imaging acquisition and preprocessing

The following 3-mm slices of MRI scans were collected from each patient: T1WI, contrast-enhanced T1W (T1Gd), ADC, DWI, and FLAIR. In the study population, preoperative MRI images were acquired within 2 weeks before surgery, and postoperative MRI images were acquired the day after surgery, whenever the patient's condition allowed. Brain MRI images were processed using a deep brain extractor to remove the skin and cranial bones [14]. The brain parenchyma was extracted based on the T1WI and DWI images (represented in gray in Fig 2).

## Tumor segmentation

To achieve semantic segmentation of the glioblastoma lesion on MRI scans, we utilized a segmentation model based on the U-net architecture, specifically designed for the segmentation of glioma [15]. Using this segmentation model, we highlighted the segmented regions, including enhanced tumors, necrosis, and cystic lesions, which are shown in red. Peritumoral edema and non-enhancing tumor areas are shown in green (Fig 2). Images with a thickness of 3 mm were used for each MRI sequence. After semantic segmentation, the tumor area, represented in red, was automatically measured. We then selected 24 segmented images per patient to ensure that the slice with the largest tumor area was included in the central part of the selected slices.

## MRI feature extraction from the latent space of a variational autoencoder

MRI features were extracted from the segmented brain images using two independently developed VAEs, as shown in Fig 3. VAE 1 was utilized to process segmented tumor lesions from

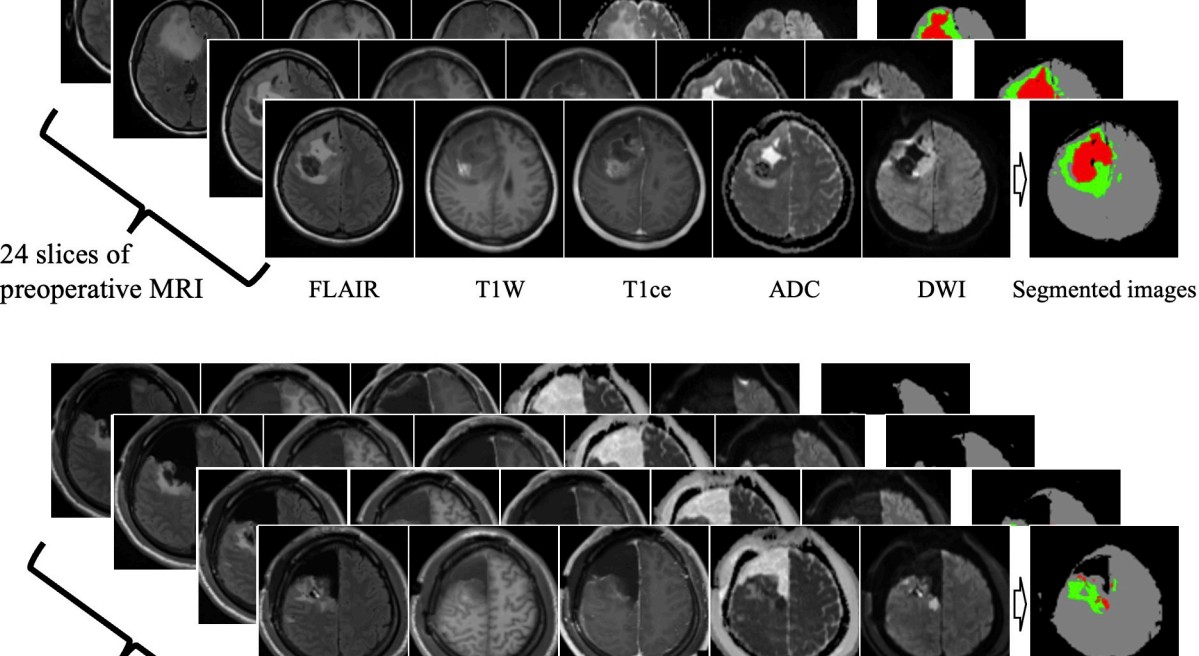

**Fig 2. Tumor segmentation using preoperative and postoperative MRI.** Using a pretrained U-net-based segmentation module, pre- and postoperative segmented images were generated from 24 slices of FLAIR, T1W, T1Gd, ADC, and DWI images. The brain parenchyma is displayed in gray, enhanced tumor lesions and necrosis in red, and peritumoral edema and non-enhancing lesions in green. T1Gd = contrast-enhanced T1W.

pre- and postoperative MRI images as input data. In contrast, VAE 2 processed data from pre- and postoperative brain parenchyma areas, serving as both input and output. The comprehensive architecture of these VAEs is demonstrated in S2 Fig. From these VAEs, MRI features were extracted from the 48-dimensional latent spaces of both VAE 1 and 2.

We pretrained these VAEs using 1,476 MRI datasets of high-grade glioma and glioblastoma MRI images from the BraTS 2020 dataset [16]. BraTS 2020 provided thin-slice MRI images of 369 patients with high-grade gliomas. The slices were extracted at regular intervals to create four different MRI datasets from a single patient. Pretraining of the VAE was conducted using annotated brain mask images and segmented tumor regions as ground truth data (Fig 1).

## Development of a KPS prediction model using neural networks

To stratify patients with a postoperative KPS score of <70 at 6 months, we developed neural network prediction models using training set (Fig 1).

These prediction models use two distinct inputs: clinical and MRI features. First, we developed a clinical neural network model using clinical parameters. In this clinical-based model, the input consists of clinical features, while the final output layer comprises two neurons, making it suitable for binary classification: predicting whether the 6-month postoperative KPS score <70 or ≥70.

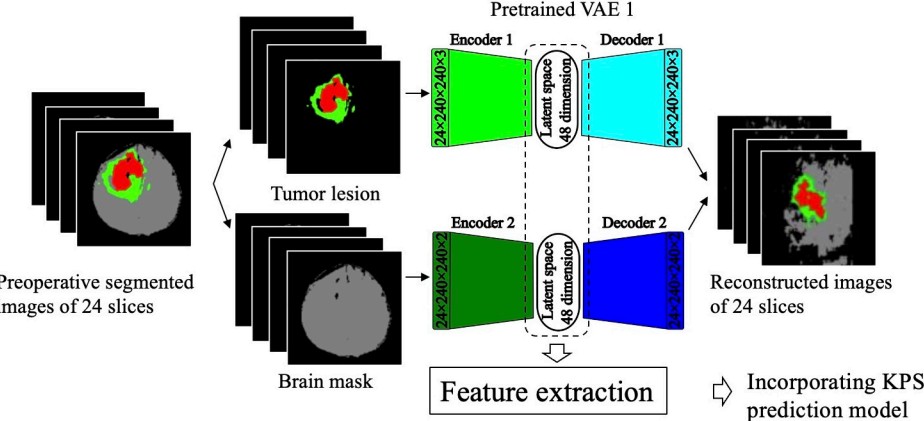

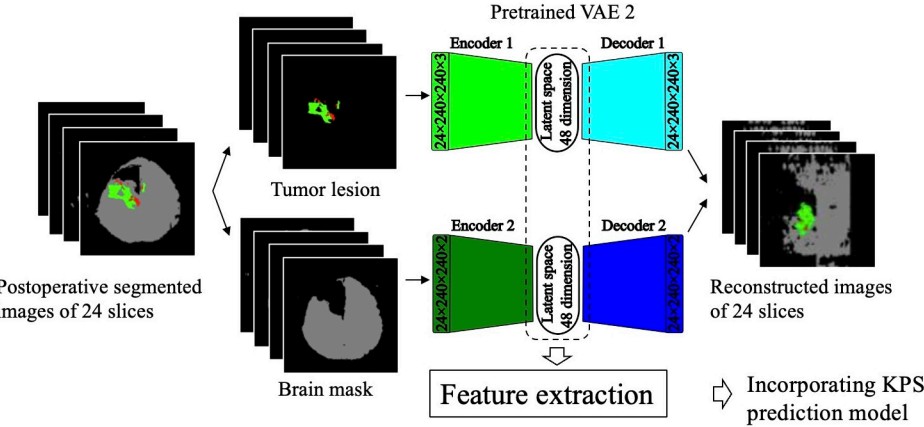

**Fig 3. Extraction of deep imaging features from the latent space of a variational autoencoder.** Twenty-four slices of the segmented images were separated into tumor lesions (red or green) and brain mask images (gray). The tumor lesion images were processed by VAE 1, where internal 3D convolutional neural networks extracted features and reduced dimensionality using encoder 1. As a result, a 48-dimensional latent space was formed, and the generated deep imaging features were incorporated into the KPS prediction model. Similarly, the brain mask images were processed by being input into VAE 2. Both VAEs 1 and 2 were pretrained using the BraTS 2020 dataset. VAE = variational autoencoder; KPS = Karnofsky performance status; BraTS = brain tumor segmentation challenge.

In addition to the clinical-based model, an MRI-based model was constructed using MRI features. Furthermore, clinical and accompanying MRI features were employed as inputs for the multimodal model. The details of the development of the prediction model are shown in S3 Fig.

During the model development with the training set, we assessed predictive performance using the area under the receiver operating characteristic curve (AUC) through 10 repetitions of fivefold cross-validation (Fig 1). This approach served as an appropriate internal validation procedure, especially given the absence of external testing [17, 18]. The effectiveness of the model was visualized through the mean ROC curves across all fivefold cross-validation. Furthermore, to evaluate the performance in fivefold cross-validation, the following metrics were computed: accuracy, sensitivity, specificity, and F1 score. The performance of the three developed models—clinical-based, MRI-based, and multimodal—was evaluated using test sets that were held out during the model development and training process. We also developed the

following machine learning models: a Random Forest classifier, XGBoost, and LightGBM. These machine learning models perform hyperparameter tuning using the GridSearch software. We extensively compared and analyzed the predictive capabilities of each classifier by considering the aforementioned metrics (S4 Fig).

## Model interpretability

Grouped permutation feature importance was utilized to determine the feature importance in the neural network-based prediction model [19]. In this analysis, features were optionally grouped into expert-defined subgroups, and a systematic assessment of their importance was conducted. In this study, during the feature extraction process, 196 MRI features were generated from preoperative and postoperative MRI imaging. To enhance interpretability, the variables derived from the preoperative brain mask image were grouped as "pre_mask variables". Similarly, variables from postoperative brain mask, preoperative tumor lesion, and postoperative tumor lesion images were grouped as "post_mask," "pre_lesion," and "post_lesion variables," respectively. Twenty-eight clinical parameters were evaluated separately, without grouping.

## Statistical analysis

All statistical analyses were performed using the SciPy library. Univariate analysis was used to examine the relationship between the 6-month postoperative KPS deterioration and clinical parameters. Fisher's exact test was used to evaluate categorical variables, whereas the Mann–Whitney U test was used for continuous variables. The cross-validated metrics of each prediction model were compared using the paired-samples t test. Statistical significance was set at P <0.05.

## Results

### Patient demographics

Of the 150 study population, 61 patients had a 6-month postoperative KPS score of <70 and 89 had ≥70. Baseline clinical parameters are presented in Table 1. Among these 150 patients, 65 (43.3%) were female, with a mean age of 64 years (range, 21–92 years). The median preoperative and 6-month postoperative KPS scores were 80 (range, 20–100) and 70 (range, 0–100), respectively. Out of the 118 patients with a preoperative KPS score of ≥70, 43 had a 6-month postoperative KPS score of <70. Conversely, among the 38 patients with a preoperative KPS score of <70, 14 had a 6-month postoperative KPS score of ≥70. These patients were divided into training and test sets. Patients who underwent surgery from December 2001 to October 2018 were assigned to the training set (n = 100), while those from November 2018 to December 2022 were included in the test set (n = 50).

### Model development

Using the training set, we developed three models to predict 6-month postoperative KPS scores of <70: a clinical-based model, an MRI-based model, and a multimodal model that incorporated both MRI features and clinical data. When both clinical and MRI data were utilized, the area under the curve (AUC) was higher than when using either clinical or MRI data alone (Fig 4A–4C and 4E). The mean AUC was 0.785 (SD 0.051) for the multimodal model using both clinical and MRI features, 0.716 (SD 0.059, P<0.001) when only clinical parameters were considered, and 0.651 (SD 0.028, P<0.001) when only MRI features were used.

**Table 1. Baseline clinical characteristics.**

| | Overall (N = 150) | KPS < 70 (n = 61) | KPS ≥ 70 (n = 89) | P value |
|---|---|---|---|---|
| Sex (female), n (%) | 65 (43.3) | 24 (39.3) | 41 (46.1) | 0.50 |
| Median age, year (range) | 64 (21–92) | 72 (32–92) | 60 (21–83) | < 0.001 |
| Dominant hand (right-handed), n (%) | 147 (98) | 58 (95.1) | 89 (100) | 0.07 |
| Preoperative epilepsy, n (%) | 40 (26.7) | 12 (19.7) | 28 (31.5) | 0.13 |
| Preoperative aphasia, n (%) | 42 (28) | 24 (39.3) | 18 (20.2) | 0.02 |
| Preoperative paralysis, n (%) | 67 (44.7) | 35 (57.4) | 32 (36.0) | 0.01 |
| Other preoperative neurological findings, n (%) | 102 (68) | 45 (73.8) | 57 (64.0) | 0.22 |
| Operation strategy (biopsy), n(%) | 27 (18) | 20 (32.8) | 7(7.9) | < 0.001 |
| Awake surgery, n (%) | 54 (36) | 19 (31.1) | 35 (39.3) | 0.39 |
| 5-aminelevulinic acid (5-ALA), n (%) | 78 (52) | 25 (41.0) | 53 (59.6) | 0.03 |
| Photodynamic therapy, n (%) | 7 (4.7) | 1 (1.6) | 6 (6.7) | 0.24 |
| Carmustine wafers placement, n(%) | 36 (24) | 9 (14.8) | 27 (30.3) | 0.03 |
| MEP or SEP monitoring, n (%) | 76 (50.1) | 26 (42.6) | 50 (56.2) | 0.14 |
| Temozolomide chemotherapy, n (%) | 142 (94.7) | 56 (91.8) | 86 (89.9) | 0.27 |
| Bevacizumab chemotherapy, n (%) | 65 (43.3) | 32 (52.5) | 33 (37.1) | 0.068 |
| MGMT promoter methylation, n (%) | 51 (34) | 19 (31.1) | 32 (36.0) | 0.60 |
| TERT promoter mutation, n (%) | 49 (32.7) | 19 (31.1) | 30 (33.7) | 0.86 |
| Median MIB-1 index, % (range) | 22.5 (0–90) | 21.2 (10–80) | 24.8 (0–90) | 0.45 |
| IHC staining of MGMT, n (%) | 54 (36) | 20 (32.8) | 34 (38.2) | 0.60 |
| Radiation dose, Gy (range) | 60 (0–63.2) | 40.05 (0–60) | 60 (0–63.2) | < 0.001 |
| Number of radiation fraction (Fr) | 30 (0–30) | 15 (0–30) | 30 (0–30) | < 0.001 |
| Tumor laterality, n (%) | | | | |
|     Right | 75 (50) | 27 (44.3) | 48 (53.9) | 0.47 |
|     Left | 63 (42) | 28 (45.9) | 35 (39.3) | - |
|     Bilateral | 12 (8.0) | 6 (9.8) | 6 (6.7) | - |
| Ependymal invasion, n (%) | 107 (71.3) | 51 (83.6) | 56 (62.9) | 0.006 |
| Midline shift, n (%) | 63 (42) | 29 (47.5) | 34 (38.2) | 0.31 |
| Corpus callosum invasion, n (%) | 41 (27.3) | 24 (39.3) | 17 (19.1) | 0.009 |
| Necrosis/cysts evident on imaging, n (%) | 138 (0.92) | 58 (95.1) | 80 (89.9) | 0.36 |
| Extent of resection, n (%) | | | | |
|     1–49% | 30 (20) | 22 (36.7) | 8 (9.0) | < 0.001 |
|     50–89% | 21 (14) | 12 (19.7) | 9 (10.1) | - |
|     90–99% | 43 (28.7) | 10 (16.4) | 33 (37.1) | - |
|     100% | 56 (37.3) | 17 (27.9) | 39 (43.8) | - |
| Karnofsky performance status | | | | |
|     Median preoperative KPS, (range) | 80 (20–100) | 70 (20–100) | 80 (40–100) | < 0.001 |
|     Preoperative KPS ≥ 70, n (%) | 118 (78.7) | 43 (70.5) | 75 (84.3) | 0.002 |
|     Median 6-months postoperative KPS, (range) | 70 (0–100) | 50 (0–60) | 80 (70–100) | < 0.001 |

Categorical variables are presented as the count of patients (percentage), while continuous variables are shown as the median value (range).

MEP, motor evoked potentials. SEP, Somatosensory evoked potentials. MGMT, O-6-Methylguanine-DNA Methyltransferase. TERT, Telomerase Reverse Transcriptase. IHC, Immunohistochemical. KPS, Karnofsky performance status.

Multimodal models also outperform clinical-based and MRI-based models in terms of accuracy (clinical parameters plus MRI features: 0.728 [SD 0.032]; clinical parameters: 0.674 [SD 0.045, P<0.021]; MRI features: 0.631 [SD 0.021, P<0.001]), sensitivity (clinical parameters plus MRI features: 0.529 [SD 0.091]; clinical parameters: 0.406 [SD 0.117, P = 0.039]; MRI

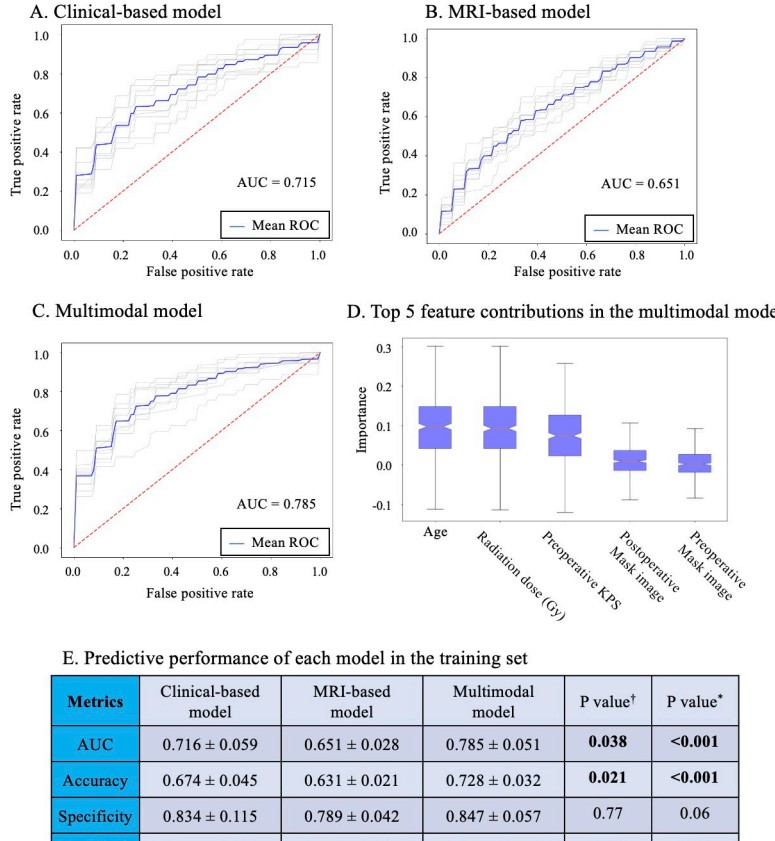

E. Predictive performance of each model in the training set

| Metrics | Clinical-based model | MRI-based model | Multimodal model | P value† | P value* |
|---|---|---|---|---|---|
| AUC | 0.716 ± 0.059 | 0.651 ± 0.028 | 0.785 ± 0.051 | **0.038** | **<0.001** |
| Accuracy | 0.674 ± 0.045 | 0.631 ± 0.021 | 0.728 ± 0.032 | **0.021** | **<0.001** |
| Specificity | 0.834 ± 0.115 | 0.789 ± 0.042 | 0.847 ± 0.057 | 0.77 | 0.06 |
| Sensitivity | 0.406 ± 0.117 | 0.402 ± 0.076 | 0.529 ± 0.091 | **0.039** | **0.01** |
| F1 score | 0.434 ± 0.086 | 0.439 ± 0.078 | 0.572 ± 0.066 | **0.002** | **0.005** |

Mean score ± Standard deviation
† Multimodal model versus clinical-based model
* Multimodal model versus MRI-based model

**Fig 4. Development of models to predict 6-month postoperative KPS score of <70 using the training set.** During model development, performance was evaluated using the training set with 10 repetitions of fivefold cross-validation. The ROC curves for each repeat of the fivefold cross-validation are represented in gray, whereas the mean ROC curve for the 10 repeats is shown in blue. The area under the curve of the model was 0.715 when using clinical parameters (**A**), 0.651 when using deep imaging features from pre- and postoperative MRI (**B**), and 0.785 when combining clinical parameters with deep imaging features (**C**). (**D**) The top five feature contributions in the multimodal model are evaluated by grouped permutation importance. (**E**) The predicted performance of each model. Data are shown as the mean score ± standard deviation. KPS = Karnofsky performance status.

features: 0.402 [SD 0.076, P = 0.01]), and F1 score (clinical parameters plus MRI features: 0.572 [SD 0.066]; clinical parameters: 0.434 [SD 0.086, P = 0.002]; MRI features: 0.439 [SD 0.078, P = 0.005]). The specificity of the multimodal model was 0.847 [SD = 0.057], which was not significantly different from the clinical-based model (0.834, [SD = 0.115, P = 0.77]) and the MRI-based model (0.789, [SD = 0.042, P = 0.06]).

The neural network-based prediction model outperformed other machine learning classifiers, including the Random Forest classifier, XGBoost, and LightGBM, when utilizing clinical parameters and MRI features. The mean AUC scores, determined through 10 repeated fivefold cross-validations with optimized hyperparameters, were calculated for each classifier. The mean AUC scores and associated P values compared with the neural network model were as follows: Random Forest classifier, 0.663 (SD 0.020, P<0.001); XGBoost, 0.705 (SD 0.029, P<0.005); and LightGBM, 0.720 (SD 0.23, P = 0.01) (S4 Fig).

| Metrics | Clinical-based model | MRI-based model | Multimodal model |
|---|---|---|---|
| AUC | 0.670 | 0.650 | 0.810 |
| Accuracy | 0.660 | 0.540 | 0.727 |
| Specificity | 0.740 | 0.815 | 0.643 |
| Sensitivity | 0.565 | 0.217 | 0.789 |
| F1 score | 0.604 | 0.303 | 0.769 |

**Fig 5. Prediction performance in the test set.** The clinical-based, MRI-based, and multimodal models were pretrained using the training set, then their prediction performance was evaluated using the test set.

Moreover, the test set evaluation of our neural network models—clinical-based, MRI-based, and multimodal—demonstrated the superiority of the multimodal model with the mean AUC of 0.810. Its performance metrics included accuracy of 0.727, specificity of 0.643, sensitivity of 0.789, and an F1 score of 0.76, mirroring the training set outcomes (Fig 5).

## Model interpretability analysis

To improve the understanding, trust, and verification of the model predictions, grouped permutation importance was applied [19]. Grouped permutation importance quantifies the feature contribution, thus providing an interpretable relationship between the incorporated features and the model prediction. Fig 4D illustrates the importance of the top five relative features for the multimodal model to predict a 6-month KPS score of <70. The most important feature in the model was "age," followed by "radiation dose (Gy)" and "preoperative KPS." Furthermore, "postoperative mask image" ranked fourth, while "preoperative mask image" ranked fifth, and these MRI features also contributed to the model prediction.

## Discussion

### Improved performance when incorporating multimodal data

The current study demonstrated that combining imaging features with clinical parameters is effective in improving the performance of clinical-based models, leading to the construction of clinically implementable models. There has been a notable increase in machine learning-based models to solve medical challenges in glioblastoma [20]; however, these models typically use data from only one modality (e.g., clinical parameters). Recently, several researchers have successfully improved the performance of models designed for clinical implementation by combining multiple modalities rather than relying on a single modality [21]. To construct a multimodal model incorporating medical imaging data into clinical-based models, it is necessary to extract imaging features from the radiographic images. The imaging features used in this process can be broadly categorized into two types: handcrafted and deep imaging features [22]. In general, handcrafted features are defined by the use of explicit formulas and are often

derived from morphological, statistical, and textural properties. On the other hand, deep imaging features are generated through a deep learning using transfer learning [23].

Lao et al. examined the importance of deep imaging and handcrafted features in the development of an overall survival prediction model for 112 patients with glioblastoma [22]. They compared 1,403 handcrafted features with 98,304 deep imaging features, which were extracted using a convolutional neural network from the preoperative MRIs. They concluded that deep imaging features contributed more to the model's performance. Recently, reviewing 69 studies of radiomic models, Demircioglu reported that clinical-based models constructed based on deep imaging features often outperform those relying on handcrafted features. Additionally, the author suggested that combining the two into a fused model could potentially enhance model performance [23]. When processing three-dimensional MRI data using a pretrained convolutional neural network, a very large number of features were generated compared to the number of patients. Therefore, strong feature selection and shrinkage are required to develop reliable clinical-based models and increase interpretation [24].

In the present study, we utilized a VAE as a feature extractor and demonstrated significant improvements in the performance of the KPS prediction model for patients with glioblastoma. Deep imaging features (i.e., MRI features) were extracted from the latent space and subsequently combined with clinical parameters. When jointly trained on data from MRI and clinical parameters in the training set, the mean AUC for predicting a 6-month postoperative KPS score of <70 was consistently higher (0.785, SD 0.051) than the models trained solely on clinical parameters (0.716, SD 0.059, P = 0.038) or MRI features (0.651, SD 0.028, P<0.001) (Fig 4E). Furthermore, in the test set, the multimodal model's AUC was 0.810, surpassing the clinical-based model's AUC of 0.670 and the MRI-based model's AUC of 0.650 (Fig 5). The important features contributing to the development of the combined model were evaluated using group permutation importance [19]. Among the top five important features, three clinical parameters, namely "age," "radiation dose," and "preoperative KPS," were included, along with the deep imaging features extracted from MRI, "postoperative mask image," and "preoperative mask image." Analysis of grouped permutation importance supported the idea that MRI features contributed to model development and improved model performance. Although MRI features had less impact on prediction performance than clinical features, we consistently observed an improvement in prediction performance when incorporating both clinical parameters and MRI features. This aligns with findings from other studies that have utilized deep learning models to merge diverse data modalities, including scenarios where clinical parameters were integrated with chest X-rays or cancer biomarkers were fused with MRI data [25, 26]. The integration of medical imaging data with corresponding medical parameters is proving to be a valuable approach for enhancing model performance.

## Clinical implication

Prediction of health status and functional impairment is critical for clinical and personal decision-making in patients with glioblastoma. A KPS score of ≥70 indicated that the patients were capable of independent self-care. Identifying patients who will require nursing or caregiving 6 months postoperatively or patients who are currently in need of care but are expected to recover independent living within 6 months is crucial for providing personalized medical management.

A low KPS was significantly correlated with a poor prognosis. For patients presenting with a KPS score of <70, less invasive treatments may be considered as an alternative to the standard protocol, which generally includes tumor removal followed by chemoradiotherapy [27]. A recent retrospective analysis revealed that the mean overall survival for patients with a

postoperative KPS score of <70 was 8 months [4]. Moreover, in clinical practice, when managing recurrent glioblastoma, their performance status can significantly influence therapeutic decision-making, which may involve options like surgical re-intervention, re-irradiation, or best supportive care [28]. Due to the limited availability of publicly accessible, precise clinical databases, research predicting the course of performance status in patients with malignant tumors is much scarcer compared to the development of prediction models for overall survival or progression-free survival. However, in recent years, several studies have reported that machine learning approaches utilizing clinical features have successfully predicted changes in performance status. Using data from patients with glioblastoma, Della Pepa et al. developed a prediction model for KPS deterioration at 6 months postoperatively, achieving an AUC of 0.81 [10]. In another report, the prediction of poor performance status in patients with systemic cancer 6 months after diagnosis achieved an AUC of 0.807 [29]. These studies utilized only clinical features and developed unimodal models. As indicated in our study, a multimodal model that incorporates both clinical and imaging features can predict performance status with significantly greater accuracy in patients with malignant tumors, including glioblastoma. The development of a KPS prediction model could help stratify patients based on their anticipated clinical course, resulting in significant implications for optimizing the balance between preserving quality of life and pursuing a more aggressive treatment approach.

The prediction of prognosis using pre- and postoperative MRI may also contribute to surgical planning when combined with generative artificial intelligence (AI) that synthesizes postoperative images. The planning and outcome of brain tumors are influenced by the surgeon's experience and involve weighing the benefits of resection against the risk of neurological impairment [30]. It has recently been reported that preoperative T1Gd MRI images can accurately predict surgical resectability using a neural network [31]. Furthermore, a recent study has demonstrated that generative AI models are capable of producing fine-quality images of pre- and postoperative brain tumors and normal parenchyma [32]. Integrating AI-generated postoperative images with a KPS prediction model might clarify optimal resection strategies and considerably enhance surgical simulations.

## Limitations

This study was limited by its relatively small patient cohort. Therefore, well-powered studies are required. As a retrospective study conducted at a single center, its external validity may have been limited by patient selection bias in our department. Multicenter external test sets, based on a prospective design, are necessary to verify the generalizability of the multimodal model. In accordance with the previously reported critical appraisal guidelines for AI research, our study aligns with Level 5B: one retrospective study with only internal data used for final performance reporting [33]. Given the rapid advancements in deep learning architectures for medical image processing, this study employed only convolutional neural networks and VAEs. Incorporating more recent architectures like vision transformers may enhance the performance of our KPS prediction model. Our understanding of surgical techniques and adjuvant therapy has gradually evolved. Patients treated in the latter years of this study likely benefited from our greater knowledge and improved treatments that are not included as clinical parameters in this study.

## Conclusions

Imaging features extracted from MRI scans using VAEs may provide valuable representations reflecting the prognosis of patients with wild-type IDH glioblastoma. The integration of these imaging features, achieved through the development of a multimodal model, significantly

enhanced the performance of the neural network-based prediction model. Predicting the 6-month postoperative KPS score has the potential to impact personalized treatment decisions, including the selection of treatment intensity and consideration of early palliative care. The future clinical implementation of the KPS prediction model offers the possibility of tailored medical interventions.

## Supporting information

**S1 Fig. Participants flow.**
(PDF)

**S2 Fig. The architecture of the variational autoencoder.**
(PDF)

**S3 Fig. The KPS prediction model development using neural network.**
(PDF)

**S4 Fig. Comparison between the neural network and other machine learning algorithm.**
(PDF)

## Author Contributions

**Conceptualization:** Tomoki Sasagasako, Yohei Mineharu.

**Data curation:** Tomoki Sasagasako, Akihiko Ueda, Yohei Mineharu.

**Formal analysis:** Tomoki Sasagasako, Akihiko Ueda, Yohei Mineharu, Yusuke Mochizuki, Souichiro Doi, Yasushi Okuno.

**Funding acquisition:** Yohei Mineharu.

**Investigation:** Tomoki Sasagasako, Yohei Mineharu, Yasushi Okuno.

**Methodology:** Tomoki Sasagasako, Akihiko Ueda, Yohei Mineharu.

**Project administration:** Akihiko Ueda, Yusuke Mochizuki, Souichiro Doi, Silsu Park, Yukinori Terada, Noritaka Sano, Masahiro Tanji.

**Resources:** Tomoki Sasagasako.

**Software:** Tomoki Sasagasako, Akihiko Ueda.

**Supervision:** Yoshiki Arakawa, Yasushi Okuno.

**Validation:** Tomoki Sasagasako, Akihiko Ueda, Yohei Mineharu.

**Visualization:** Tomoki Sasagasako, Akihiko Ueda.

**Writing – original draft:** Tomoki Sasagasako.

**Writing – review & editing:** Yohei Mineharu, Yoshiki Arakawa, Yasushi Okuno.

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
