## [Decision Letter · Decision Letter 0]

13 Sep 2024

PONE-D-24-13237Postoperative Karnofsky performance status prediction in patients with IDH wild-type glioblastoma: a multimodal approach integrating clinical and deep imaging featuresPLOS ONE

Dear Dr. Mineharu,

Thank you for submitting your manuscript to PLOS ONE. After careful consideration, we feel that it has merit but does not fully meet PLOS ONE’s publication criteria as it currently stands. Therefore, we invite you to submit a revised version of the manuscript that addresses the points raised during the review process.

We look forward to receiving your revised manuscript.

Kind regards,

Kevin Camphausen

Academic Editor

PLOS ONE

3. Thank you for stating the following financial disclosure: [This work was supported by the Ministry of Education, Culture, Sports, Science, and Technology (MEXT) under the RIKEN joint research and collaboration fund for “Translational Research in Basic and Clinical Sciences for the Construction of an AI Pharmaceutical Platform”.]. Please state what role the funders took in the study. If the funders had no role, please state: "The funders had no role in study design, data collection and analysis, decision to publish, or preparation of the manuscript." If this statement is not correct you must amend it as needed. Please include this amended Role of Funder statement in your cover letter; we will change the online submission form on your behalf.

4. We note that you have indicated that there are restrictions to data sharing for this study. For studies involving human research participant data or other sensitive data, we encourage authors to share de-identified or anonymized data. However, when data cannot be publicly shared for ethical reasons, we allow authors to make their data sets available upon request. For information on unacceptable data access restrictions, please see http://journals.plos.org/plosone/s/data-availability#loc-unacceptable-data-access-restrictions. Before we proceed with your manuscript, please address the following prompts: a) If there are ethical or legal restrictions on sharing a de-identified data set, please explain them in detail (e.g., data contain potentially identifying or sensitive patient information, data are owned by a third-party organization, etc.) and who has imposed them (e.g., a Research Ethics Committee or Institutional Review Board, etc.). Please also provide contact information for a data access committee, ethics committee, or other institutional body to which data requests may be sent. b) If there are no restrictions, please upload the minimal anonymized data set necessary to replicate your study findings to a stable, public repository and provide us with the relevant URLs, DOIs, or accession numbers. Please see http://www.bmj.com/content/340/bmj.c181.long for guidelines on how to de-identify and prepare clinical data for publication. For a list of recommended repositories, please see https://journals.plos.org/plosone/s/recommended-repositories. You also have the option of uploading the data as Supporting Information files, but we would recommend depositing data directly to a data repository if possible. Please update your Data Availability statement in the submission form accordingly.

5. In the online submission form, you indicated that [The clinical data in this study cannot be shared publicly because it contains potentially identifying or sensitive patient information. However, the clinical data are available, on reasonable request, from the corresponding author.]. All PLOS journals now require all data underlying the findings described in their manuscript to be freely available to other researchers, either 1. In a public repository, 2. Within the manuscript itself, or 3. Uploaded as supplementary information. This policy applies to all data except where public deposition would breach compliance with the protocol approved by your research ethics board. If your data cannot be made publicly available for ethical or legal reasons (e.g., public availability would compromise patient privacy), please explain your reasons on resubmission and your exemption request will be escalated for approval.

Additional Editor Comments (if provided):

Reviewers' comments:

Reviewer's Responses to Questions

**Comments to the Author**

1. Is the manuscript technically sound, and do the data support the conclusions?

Reviewer #1: Yes

Reviewer #2: Yes

2. Has the statistical analysis been performed appropriately and rigorously? 

Reviewer #1: I Don't Know

Reviewer #2: Yes

3. Have the authors made all data underlying the findings in their manuscript fully available?

Reviewer #1: No

Reviewer #2: Yes

4. Is the manuscript presented in an intelligible fashion and written in standard English?

Reviewer #1: Yes

Reviewer #2: Yes

5. Review Comments to the Author

Reviewer #1: This is very well written paper and the rationale is well thought out. I enjoyed reading this paper.

I have only very minor comments.

The paper would be enhanced by 1) discussing the limitations of the methodology, 2) the prospects for validation and 3) by comparing this analysis and its results with other similar studies.

Reviewer #2: A manuscript entitled “Postoperative Karnofsky performance status prediction in patients with IDH wild-type glioblastoma: a multimodal 3 approach integrating clinical and deep imaging features” is submitted. Authors intended to develop a 6-month postoperative KPS prediction model by combining clinical data with deep learning-based image features from pre- and postoperative MRI scans.

Overall, the study presents the data of original research. Conclusions are largely supported by the data.

A few comments as below:

1.Please consider revising the statement in line 67-69, “The 6-month postoperative KPS was chosen for its clinical practicality, as it aligns with the standard timeframe for evaluating disease progression after a typical course of adjuvant treatments.” Most glioblastoma patients are in the middle of adjuvant treatment.

2. It would be important to address how the pre-surgical, intra-surgical variables and other clinical factors were selected.

3. It is unclear why patients were assigned into training and test sets by time range instead of randomly.

4. Despite the robust MRI-based model building utilizing the VAE, the multimodal model only improves the predictability slightly compared to clinical-only model.

5. It is unclear how could the KPS prediction model contribute to the surgical planning, as the modeling requires pre- and post-operative MRIs.

6. PLOS authors have the option to publish the peer review history of their article (what does this mean?). If published, this will include your full peer review and any attached files.

Reviewer #1: No

Reviewer #2: No

---

## [Author Response · Author response to Decision Letter 0]

3 Oct 2024

Detailed point-by-point responses to the Reviewers

October 3, 2024

Thank you very much for your invaluable advice. We want to thank the editor and reviewers for their effort and time in reviewing our manuscript. We have carefully considered each comment. We fundamentally agree with all the comments and have incorporated the suggestions into the revised manuscript. The responses to the reviewers and the changes made in the revised manuscript are shown below.

Response to Reviewer #1

Comment 1:

 " This is very well written paper and the rationale is well thought out. I enjoyed reading this paper. I have only very minor comments. The paper would be enhanced by 1) discussing the limitations of the methodology, 2) the prospects for validation and 3) by comparing this analysis and its results with other similar studies."

Author's response:

We really appreciate your careful review and kind comments. We have revised the Discussion and Limitations sections according to your suggestions.

1) Discussing the limitations of the methodology

One of the main limitations in the development of our prediction model relates to the extraction of MRI features. Given the rapid advancements in deep learning architectures for medical image processing, this study employed only convolutional neural networks (CNNs) and variational autoencoders (VAEs). Incorporating more recent architectures like vision transformers can potentially enhance the performance of the KPS prediction model.

2) The prospects for validation

The nature of the retrospective-single center analysis leads to potential selection bias. Multicenter external test sets, based on a prospective design, are necessary to verify the generalizability of the model. 

3) Comparing this analysis and its results with other similar studies

Research predicting the course of performance status in patients with malignant tumors is limited, in part due to the scarce availability of publicly accessible, precise clinical databases. However, in recent years, several studies have successfully employed machine learning approaches using clinical features to predict changes in performance status. These studies utilized only clinical features and developed unimodal models. As indicated in our study, a multimodal model that incorporates both clinical and imaging features can predict the performance status with significant accuracy in patients with malignant tumors, including glioblastoma.

Changes to the text:

“Multicenter external test sets, based on a prospective design, are necessary to verify the generalizability of the multimodal model.” (Limitations: page 18–19, lines 387–388)

“Given the rapid advancements in deep learning architectures for medical image processing, this study employed only convolutional neural networks and VAEs. Incorporating more recent architectures like vision transformers may enhance the performance of our KPS prediction model.” (Limitations: page 19, lines 390–393)

“However, in recent years, several studies have reported that machine learning approaches utilizing clinical features have successfully predicted changes in performance status. Using data from patients with glioblastoma, Della Pepa et al. developed a prediction model for KPS deterioration at 6 months postoperatively, achieving an AUC of 0.81 [10]. In another report, the prediction of poor performance status in patients with systemic cancer 6 months after diagnosis achieved an AUC of 0.807 [29]. These studies utilized only clinical features and developed unimodal models. As indicated in our study, a multimodal model that incorporates both clinical and imaging features can predict performance status with significantly greater accuracy in patients with malignant tumors, including glioblastoma.” (Discussion: page 18, lines 363–371)

Response to Reviewer #2

Comment 1:

 " Please consider revising the statement in line 67-69, “The 6-month postoperative KPS was chosen for its clinical practicality, as it aligns with the standard timeframe for evaluating disease progression after a typical course of adjuvant treatments.” Most glioblastoma patients are in the middle of adjuvant treatment. "

Author’s response:

We are grateful for your thoughtful review. We fully agree that many patients are in the middle of adjuvant therapy at the 6-month postoperative mark in the standard protocol for glioblastoma. Our original statement was indeed misleading. We revised this sentence as follows.

Changes to the text:

“Commonly, the 6-month postoperative KPS score is used for evaluating disease progression in glioblastoma clinical trials [5].” (Introduction: page 4, lines 63–64)

Comment 2:

 " It would be important to address how the pre-surgical, intra-surgical variables and other clinical factors were selected."

Author’s response:

We appreciate your crucial comment. The pre-surgical, intra-surgical, and other clinical parameters used in our study are based on similar variables used by previous machine learning research on prediction models of glioblastoma prognosis [1,2]. These parameters are routinely assessed and available in clinical practice.

Changes to the text:

“According to previous research on machine-learning-based models predicting glioblastoma prognosis, the following 28 variables were incorporated as clinical parameters [10,11].” (Materials and methods: page 6, lines 115–116)

Comment 3:

 " It is unclear why patients were assigned into training and test sets by time range instead of randomly. "

Author’s response:

Thank you for your insightful comment. In studies with relatively small populations, the performance of a model on a test set can vary significantly with each random split, largely due to substantial changes in the distribution of the test set data [3]. Our study, with a cohort of 150 cases, may experience fluctuations in model performance that could either overestimate or underestimate the true performance, depending on the distribution of the test set in each random split. To mitigate this randomness and prevent the bias of arbitrarily selecting the test set through repeated random splits, we chose to split the cases based on the time of surgery into training and test sets in a 2:1 ratio. This method of splitting has been frequently employed in previous studies with similar cohort sizes to ensure reliability in the evaluation process [4,5].

Changes to the text:

“In studies with relatively small populations, the performance of a model on a test set can significantly vary with each random split due to substantial changes in the distribution of the test set data [7]. To mitigate the bias associated with arbitrarily selecting the test set through repeated random splits, we chose to divide the cases based on their time range into training and test sets in a 2:1 ratio, a method frequently employed in prior studies with similar cohort sizes [8,9].” (Materials and methods: page 6, lines 106–111)

Comment 4:

 " Despite the robust MRI-based model building utilizing the VAE, the multimodal model only improves the predictability slightly compared to clinical-only model."

Author’s response:

We appreciate your insightful comment. Indeed, the MRI-only model exhibited inferior performance compared to the clinical-only model in the training set (Figs 4A, B). Consequently, incorporating MRI features into the clinical variables slightly improved prediction performance relative to the clinical-only model (Fig 4E). Additionally, the grouped permutation feature importance analysis revealed that the top three variables contributing to the performance of the multimodal model were clinical features (Fig 4D). Although our findings suggest that imaging features have a lesser impact than clinical features on the performance of predictive models for glioblastoma patients, the integration of clinical parameters with MRI features consistently improved prediction accuracy. These results highlight the potential benefits of enhancing the clinical-based model with imaging features to improve prediction performance.

Changes to the text:

“Although MRI features had less impact on prediction performance than clinical features, we consistently observed an improvement in prediction performance when incorporating both clinical parameters and MRI features.” (Discussion: page 17, lines 339–342)

Comment 5:

 " It is unclear how could the KPS prediction model contribute to the surgical planning, as the modeling requires pre- and post-operative MRIs."

Author's response

We are grateful for your invaluable comment. We apologize for any ambiguity in our initial discussion. We anticipate a scenario where AI-generated postoperative images, rather than actual images, are integrated into a KPS prediction model for surgical planning. Recent research has successfully developed surgical resectability prediction models and generative models that produce high-quality images of both pre- and postoperative MRIs [6–8]. Integrating AI-generated postoperative images into a KPS prediction model may clarify optimal resection strategies and significantly enhance surgical simulations, thereby improving patient outcomes.

Changes to the text:

“The prediction of prognosis using pre- and postoperative MRI may also contribute to surgical planning when combined with generative artificial intelligence (AI) that synthesizes postoperative images.” (Discussion: page 18, lines 375–377)

“Furthermore, a recent study has demonstrated that generative AI models are capable of producing fine-quality images of pre- and postoperative brain tumors and normal parenchyma [32]. Integrating AI-generated postoperative images with a KPS prediction model might clarify optimal resection strategies and considerably enhance surgical simulations.” (Discussion: page 18, lines 380–383)

References

1. Della Pepa GM, Caccavella VM, Menna G, Ius T, Auricchio AM, Chiesa S, et al. Machine learning–based prediction of 6-month postoperative Karnofsky performance status in patients with glioblastoma: capturing the real-life interaction of multiple clinical and oncologic factors. World Neurosurg. 2021;149: e866–e876. doi:10.1016/j.wneu.2021.01.082

2. Della Pepa GM, Caccavella VM, Menna G, Ius T, Auricchio AM, Sabatino G, et al. Machine learning-based prediction of early recurrence in glioblastoma patients: a glance towards precision medicine. Neurosurgery. 2021;89: 873–883. doi:10.1093/neuros/nyab320

3. Xu Y, Goodacre R. On splitting training and validation set: a comparative study of cross-validation, bootstrap and systematic sampling for estimating the generalization performance of supervised learning. J Anal Test. 2018;2: 249–262. doi:10.1007/s41664-018-0068-2

4. Park JE, Kim HS, Lee J, Cheong EN, Shin I, Ahn SS, et al. Deep-learned time-signal intensity pattern analysis using an autoencoder captures magnetic resonance perfusion heterogeneity for brain tumor differentiation. Sci Rep. 2020;10: 1–11. doi:10.1038/s41598-020-78485-x

5. Yun J, Yun S, Park JE, Cheong EN, Park SY, Kim N, et al. Deep learning of time-signal intensity curves from dynamic susceptibility contrast imaging enables tissue labeling and prediction of survival in glioblastoma. AJNR Am J Neuroradiol. 2023;44: 543–552. doi:10.3174/ajnr.A7853

6. Miao X, Chen H, Tang M, Huang D, Gao T, Chen Y. Post-operative MRI synthesis from pre-operative MRI and post-operative CT using conditional GAN for the assessment of degree of resection. Displays. 2024;83: 102742. doi:10.1016/j.displa.2024.102742

7. Marcus AP, Marcus HJ, Camp SJ, Nandi D, Kitchen N, Thorne L. Improved prediction of surgical resectability in patients with glioblastoma using an artificial neural network. Sci Rep. 2020;10: 1–9. doi:10.1038/s41598-020-62160-2

8. Mukherkjee D, Saha P, Kaplun D, Sinitca A, Sarkar R. Brain tumor image generation using an aggregation of GAN models with style transfer. Sci Rep. 2022;12: 1–17. doi:10.1038/s41598-022-12646-y

#########################

Response to the Specific Journal Requirements

Dear Editor,

Thank you very much for the opportunity to revise and resubmit our manuscript. We sincerely appreciate the journal's support and efforts throughout the review process.

We have addressed the specific requirements requested by the journal as follows:

1. We reviewed our manuscript against the PLOS ONE style templates and ensured full adherence to the formatting guidelines, including proper file naming conventions.

2. In compliance with PLOS ONE's code-sharing policy, we have made all author-generated code available via GitHub at the following URL: https://github.com/TomokiSasagasako/GBM_KPS_prediction.git. This link has been included in the manuscript to ensure accessibility and facilitate reproducibility.

3. A financial disclosure statement has been added to the manuscript, clarifying:

"The funders had no role in study design, data collection and analysis, decision to publish, or preparation of the manuscript." (page 20, lines 428–430)

4&5. Regarding the request for data publication, we consulted with our Research Ethics Committee. Unfortunately, due to the sensitive nature of the clinical data, which includes potentially identifiable information (such as the name of the hospital, diagnostic details, age, gender, and clinical outcomes), we are unable to provide unrestricted access. Sharing such information could lead to the identification of individual patients, and therefore it is not permissible under our ethical guidelines. This decision is consistent with the ethical guidance provided by the previously referenced article: http://www.bmj.com/content/340/bmj.c181.long.

For documentation of data availability, we have updated our manuscript to include the following statement:

" Data availability

The code used to develop the model described herein is publicly available on GitHub: https://github.com/TomokiSasagasako/GBM_KPS_prediction.git

The raw data in this study cannot be shared publicly as it includes identifiable information such as the name of the hospital (Kyoto University Hospital), diagnostic details, age, gender, and clinical outcomes. The raw data are available upon reasonable request from the corresponding author or Kyoto University Graduate School and Faculty of Medicine, Ethics Committee via email (ethcom@kuhp.kyoto-u.ac.jp) or telephone (+81-75-753-4680)." (page 20, lines 416–423)

Thank you for your consideration. I look forward to your response.

Sicerely,

Yohei Mineharu

---

## [Editor Report · Decision Letter 1]

7 Oct 2024

Postoperative Karnofsky performance status prediction in patients with IDH wild-type glioblastoma: a multimodal approach integrating clinical and deep imaging features

PONE-D-24-13237R1

Dear Dr. Mineharu,

We’re pleased to inform you that your manuscript has been judged scientifically suitable for publication and will be formally accepted for publication once it meets all outstanding technical requirements.

Kind regards,

Kevin Camphausen

Academic Editor

PLOS ONE

Additional Editor Comments (optional):

Thank you
---

## [Editor Report · Acceptance letter]

10 Oct 2024

PONE-D-24-13237R1 

PLOS ONE

Dear Dr. Mineharu, 

I'm pleased to inform you that your manuscript has been deemed suitable for publication in PLOS ONE. Congratulations! Your manuscript is now being handed over to our production team.

Kind regards, 

on behalf of

Dr. Kevin Camphausen 

Academic Editor

PLOS ONE